# Temporal structure in associative retrieval

Zeb Kurth-Nelson[1,2]*, Gareth Barnes[1], Dino Sejdinovic[3], Ray Dolan[1,2], Peter Dayan[3]

[1]Wellcome Trust Centre for Neuroimaging, University College London, London, United Kingdom; [2]Max Planck UCL Centre for Computational Psychiatry and Ageing Research, University College London, London, United Kingdom; [3]Gatsby Computational Neuroscience Unit, University College London, London, United Kingdom

**Abstract** Electrophysiological data disclose rich dynamics in patterns of neural activity evoked by sensory objects. Retrieving objects from memory reinstates components of this activity. In humans, the temporal structure of this retrieved activity remains largely unexplored, and here we address this gap using the spatiotemporal precision of magnetoencephalography (MEG). In a sensory preconditioning paradigm, 'indirect' objects were paired with 'direct' objects to form associative links, and the latter were then paired with rewards. Using multivariate analysis methods we examined the short-time evolution of neural representations of indirect objects retrieved during reward-learning about direct objects. We found two components of the evoked representation of the indirect stimulus, 200 ms apart. The strength of retrieval of one, but not the other, representational component correlated with generalization of reward learning from direct to indirect stimuli. We suggest the temporal structure within retrieved neural representations may be key to their function.

## Introduction

Associative memory in animals and humans provides a model of the environment. Retrieval of such memories, driven by cues or occurring autonomously, is suggested as central to a wide variety of processes and functions, including online and offline planning and model-learning (*Sutton, 1991*; *Moore and Atkeson, 1993*; *Foster and Wilson, 2006*; *Johnson and Redish, 2007*; *Hasselmo, 2008*; *Lisman and Redish, 2009*; *Gupta et al., 2010*; *van der Meer et al., 2010*; *Jadhav et al., 2012*; *Wimmer and Shohamy, 2012*; *Pfeiffer and Foster, 2013*; *Singer et al., 2013*), cognitive search (*Kurth-Nelson et al., 2012*; *Todd et al., 2012*; *Morton et al., 2013*), mental time travel (*Hopfield, 2010*; *Schacter et al., 2012*), memory maintenance and consolidation (*Marr, 1971*; *Nádasdy et al., 1999*; *Káli and Dayan, 2004*; *Kuhl et al., 2012*; *Deuker et al., 2013*) as well as temporal expectation (*Sakai and Miyashita, 1991*; *Rainer et al., 1999*).

Retrieval is classically linked to reinstantiation of a particular distributed spatial pattern of neural activity mirroring that evoked by the original experience of the object or context being retrieved (*Tulving and Thomson, 1973*; *Nyberg et al., 2000*; *Hoffman and McNaughton, 2002*; *Polyn et al., 2005*; *Johnson and Rugg, 2007*; *Gelbard-Sagiv et al., 2008*; *Danker and Anderson, 2010*; *Rissman and Wagner, 2012*; *Miller et al., 2013*; *Kuhl and Chun, 2014*). However, electrophysiology experiments robustly demonstrate that when an object is directly experienced, the evoked pattern of neural activity evolves rapidly over tens to hundreds of milliseconds (*Makeig et al., 1997*; *Schmolesky et al., 1998, 1998*; *Näätänen and Winkler, 1999*; *VanRullen and Thorpe, 2001*; *Rossion and Jacques, 2008*; *Schneider et al., 2008*; *Cichy et al., 2014*). This implies that direct experience of an object evokes multiple distinct spatial patterns of neural activity in sequence. However, these distinct spatial patterns have never been identified independently at retrieval.

*For correspondence:
z.kurth-nelson@ucl.ac.uk

Competing interests: The authors declare that no competing interests exist.

**eLife digest** Seeing an object triggers a complex and carefully orchestrated dance of brain activity. The spatial pattern of the brain activity encoding the object can change multiple times even within the first second of seeing the object. These rapid changes appear to be a core feature of how the brain understands and processes objects.

Yet little is known about how these patterns unfold through time when we *remember* an object. Remembering, or retrieving information about objects, is how we use our knowledge of the world to make good decisions. It is not clear whether, during remembering, there are rapid changes in the patterns similar to those that happen when directly seeing an object. Mapping brain activity during remembering could help us understand how stored information can guide decisions.

Using recently developed methods in brain imaging and statistics, Kurth-Nelson et al. found that two distinct patterns of brain activity appeared when viewing particular objects. One occurred around 200 milliseconds after viewing an object, and the other appeared a bit later, by about 400 milliseconds. Later, when remembering the object, these patterns reappeared in the brain, but at different points in time. Furthermore, these two patterns had distinct roles in learning associated with the objects to guide later decisions.

This work shows that rapid changes in the pattern of neuronal activity are central to how stored information is retrieved and used to make decisions.

At retrieval, recent studies have explored the fast evolution of neural representation. EEG studies provide evidence that some information is retrieved as early as 300 ms following a cue (e.g., *Johnson et al., 2008*; *Yick and Wilding, 2008*; *Wimber et al., 2012*). *Manning et al. (2011)*, (*2012*), using electrocorticography, and *Jafarpour et al. (2014)*, using MEG, showed that oscillatory patterns are also reinstated during retrieval; in two of these studies the predominance of low oscillatory frequencies in reinstatement suggests a potential spectral signature.

However, the dynamics of representation during direct experience of an object have never been tied to the dynamics of retrieval. It is not known which of the patterns evoked in sequence by direct experience are reinstantiated during retrieval, what the temporal relationship is in their retrieval, or what functional significance this has.

Recent advances in multivariate methods for MEG have greatly improved our ability to discern fast-changing distributed representations in humans (*Carlson et al., 2013*; *Cichy et al., 2014*; *Jafarpour et al., 2013*, *2014*; *van de Nieuwenhuijzen et al., 2013*; *Sandberg et al., 2013*). Here, we apply these methods to a simple sensory preconditioning task adapted from *Wimmer and Shohamy (2012)*. Sensory preconditioning is a well-established paradigm in which subjects first form an association between two stimuli ('direct' or $S_d$ and 'indirect' or $S_i$) and then form an association between the direct stimulus and a reward (*Brogden, 1939*). Generalization of value to the indirect stimulus is evidence of retrieving the learned association (*Gewirtz and Davis, 2000*). Using fMRI, *Wimmer and Shohamy (2012)* showed that neural representations of the associated indirect stimulus are reinstated when direct stimuli are presented during the Reward-learning phase, and this retrieval is linked to the generalization of value from direct to indirect stimuli. This suggests that reinstatement through the learned associative link may be part of the mechanism for value updating. Our aim here is to explore the temporal structure of this reinstatement, which may help to shed light on the mechanisms of value updating as well as providing general insight into the dynamics of representations during retrieval.

We therefore examined retrieval in the same paradigm, using MEG to gain temporal precision. We show that the neural representation of the indirect stimulus can be decomposed into at least two temporal components with distinct properties, and these are retrieved at different times during the Reward-learning phase. The retrieval of only one of these components is correlated with a behavioral measure of the generalization of value across the learned associations.

## Results

### Behavior

We used a slightly modified version of the behavioral task employed by *Wimmer and Shohamy (2012)*. This involved three phases (*Figure 1A*). In the Association phase, subjects watched visual stimuli

appearing sequentially at the center of the screen. The stimuli alternated between photographs ('$S_i$') and circular fractals ('$S_d$'), with a short blank fixation interval between each stimulus. Each $S_i$ came from one of three categories (face/body/scene), and each unique $S_i$ was deterministically followed by a unique $S_d$, thus establishing a pairing between $S_i$ and $S_d$ images. As in *Wimmer and Shohamy (2012)*, debriefing revealed that subjects were not aware of the $S_i$–$S_d$ pairings. There were two unique $S_i$ in each category; making for a total of six unique $S_i$ and six unique $S_d$ stimuli used for later phases (along with six additional unique $S_i$ and six additional unique $S_d$ that functioned as dummies for the Association phase cover task and were included in the imaging analysis).

In the Reward phase, of the two $S_d$ images associated with a category of $S_i$ images, one, which we therefore call $S_d$+ was followed by a reward on 14 out of 18 presentations (and otherwise by a neutral outcome, a blue square); the other, which we call $S_d$- was always followed by a neutral outcome. By virtue of the prior pairing, this established an $S_i$+ and $S_i$− for each category.

In the Decision phase, subjects were faced with pairwise choices between an $S_i$+ and an $S_i$−, or an $S_d$+ and an $S_d$−. The two items always had the same category (face/body/scene) for $S_i$, or associated category for $S_d$. Subjects exhibited a strong preference for $S_d$+ over $S_d$− (p = $6.9 \times 10^{-4}$), but as a group showed no evidence of preferring $S_i$+ over $S_i$− (p = 0.9) (*Figure 1B*).

## Distinct temporal components of neural object representation

Neural activity was recorded by magnetoencephalography (MEG) during all three phases. We first explored where in space and time the MEG signal carried information about the $S_i$ stimuli being presented in the Association phase. Using one-way ANOVA, we found that the raw amplitude, in single time bins, of the event-related field (ERF) at many individual sensors was significantly related to the $S_i$ category (*Figure 2*). (The significance threshold was set to 95% of peak-level over space and time from 100 random category label shuffles, to correct conservatively for multiple comparisons.)

Next, we built a multivariate linear SVM classifier, which combined the reports of multiple sensors (*Figure 3A*). As in many previous studies (cf. *Norman et al., 2006*; *Cichy et al., 2014*), the extra sensitivity achieved by combining multiple features supported the use of multivariate analysis to track neural representations (*Figure 3—figure supplement 1*). We constructed null distributions at each time bin by repeating this procedure 100 times with randomly shuffled category labels. At 200 ms post-stimulus, the 95th percentile of the null distribution was 35.0% accuracy, and the median was 33.7% (deviating from 1/3rd only due to the finite number of shuffles).

We observed two distinct peaks in multivariate classification performance, one centered approximately around 200 ms and the other around 400 ms post-stimulus onset. Although these peaks had measurable width, for simplicity, we will henceforth refer to them as '200 ms' and '400 ms'. To test more formally for two distinct peaks in classification, we asked whether there was significant concavity in the evolving classification accuracy in the interval from 200 to 400 ms, by regressing the classification accuracy against linear and quadratic functions of time. At the group level, the quadratic term was significantly different from zero (p = 0.02). We also performed this regression on the accuracy curves from individual subjects; many subjects trended toward a positive quadratic term, but none reached significance at a Bonferroni-corrected threshold (*Figure 3—figure supplement 2*). Finally, to rule out any peculiarities in the SVM algorithm being responsible for two distinct peaks in classification accuracy, we also repeated the same analysis at the group level with a variety of nearest-mean classifiers and found the same pattern (*Figure 3—figure supplement 3*).

Given past observations and ideas about separate post-stimulus phases encoding qualitatively different kinds of stimulus information (*Schmolesky et al., 1998*; *Lamme and Roelfsema, 2000*; *Riesenhuber and Poggio, 2000*; *Engel et al., 2001*; *Bar, 2003*; *Cichy et al., 2014*), we asked if these two peaks had different representational similarity structure. We calculated representation similarity matrices (*Kriegeskorte et al., 2008*), which reflect the similarity in activation patterns between each pair of unique stimuli. We found that at 200 ms, the activity patterns evoked by stimuli within a category were no more similar than those evoked by stimuli in different categories (*Figure 3B*, left panel; p = 0.2, paired t-test between subjects); whereas at 400 ms, patterns within a category were substantially more similar than between categories (*Figure 3B*, right panel; p = $5 \times 10^{-7}$). This is consistent with the idea that the dominant coding of stimulus information changes between 200 and 400 ms.

Further supporting the idea that the later component of the ERF had a relatively more dominant coding of categorical information, we found that the cross-validated performance of a linear

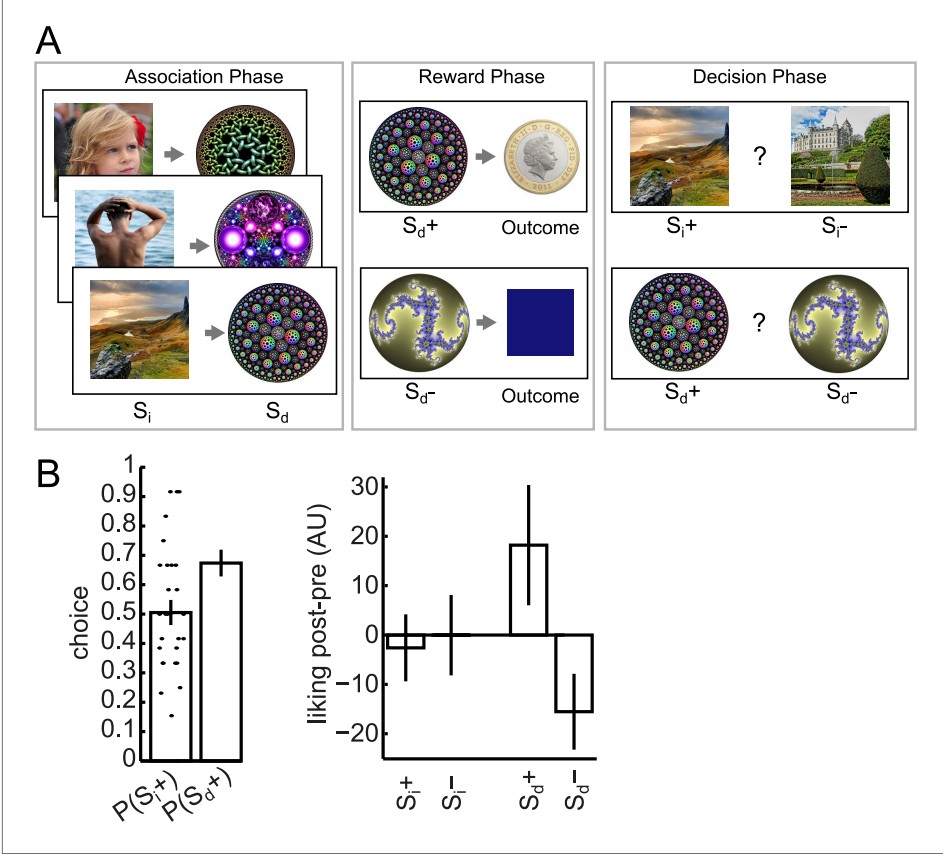

**Figure 1**. Task design and behavior. Subjects participated in a sensory preconditioning task comprising three phases: Association, Reward and Decision. (**A**) In the Association phase, subjects were exposed to pairs of stimuli (presented sequentially). One member (called $S_i$) of each pair was taken from one of three classes (faces, bodies, and scenes); the other member ($S_d$) was a fractal. In the Reward phase, some of the fractals (labelled $S_d+$) were paired with reward; the others (labelled $S_d-$) were not. Through the pairing, this implicitly established a separation between $S_i+$ and $S_i-$. In the Decision phase, subjects chose between $S_i+$ and $S_i-$ within the same category, or between $S_d+$ and $S_d-$. All photos shown are from pixabay.com and are in the public domain. (**B**) In the Decision phase, subjects displayed a strong preference for $S_d+$ over $S_d-$ ($p = 6.9 \times 10^{-4}$, one-sample t-test). There was no preference at the group level for $S_i+$ over $S_i-$, but we exploited the variability between subjects for value-related analyses. The change in relative liking from before to after the experiment was more positive for $S_d+$ than $S_d-$ ($p = 0.04$, one-sample t-test); but there was no significant difference between the changes for $S_i+$ and $S_i-$. Bar heights show group means and dots show individual subjects. Error bars show standard error of the mean.

SVM in a 6-way discrimination of fractal identity was sharply peaked at 160 ms post-stimulus onset, and lacked a substantial second peak (*Figure 3C*). We note that a shift in the timing of the early peak from ~200 ms to ~160 ms could be consistent with previous observations (*Bobak et al., 1987*; *Cichy et al., 2014*) that the precise timing of each wave of representation is sensitive to the particular stimuli concerned.

## $S_d$ elicits retrieval of associated $S_i$ representation

During the Reward phase, $S_d$ (fractals) and outcomes (coin/blue square) were presented. We confirmed it was possible to predict the identity of fractals (cf. *Figure 3C*) and outcomes (*Figure 3—figure supplement 4*) reliably based on the MEG signal. However, the main intention of our study was to examine whether the activity evoked by these stimuli contained information about the $S_i$ stimulus with which the $S_d$ had been associated. To this end, we trained classifiers on neural responses to $S_i$ in the Association phase (exactly as above, but using all trials because cross-validation was not necessary), and tested these classifiers on neural responses elicited in the Reward phase when $S_d$ was presented. The classifier

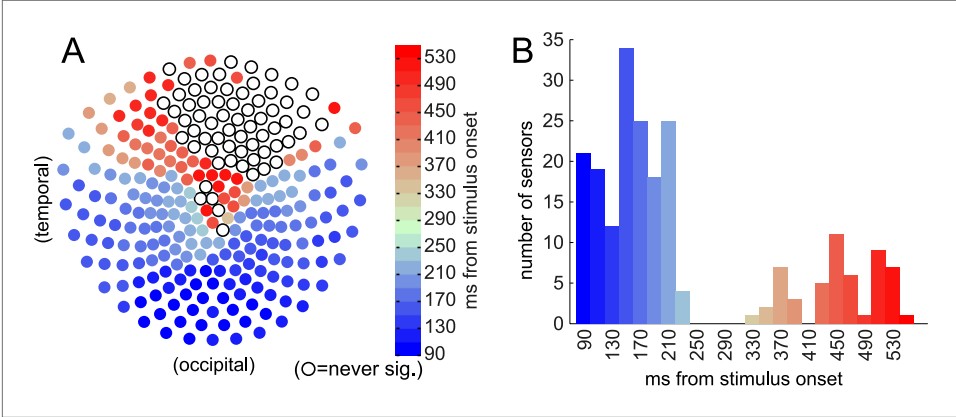

**Figure 2**. Event-related field (ERF) discriminates between categories (face/body/scene) at time of $S_i$ presentation. Sensors became category-discriminative in two waves. (**A**) The first time, relative to stimulus onset, when the relationship between ERF amplitude and category membership became significant by ANOVA (significance threshold set at 95% of peak-level (across all sensors and all time) $\log_{10}(p)$ of 100 shuffles) at each of 275 sensors. Many occipital and temporal sensors first became predictive of $S_i$ category between 90 and 230 ms post stimulus onset, followed by some parietal and frontal sensors ranging from 330–550 ms post stimulus onset. Open circles indicate the sensors that never reached 95% peak-level. (**B**) Histogram of how many sensors first became significantly discriminative at each time following stimulus presentation.

was considered to be correct if it reported the category label of the $S_i$ that had previously been paired with this $S_d$. We performed this train-on-$S_i$, test-on-$S_d$ procedure for every pair of times relative to the onsets of $S_i$ (in the Association phase) and $S_d$ (in the Reward phase), leading to a 2-D grid of classification accuracies (*Figure 4A*). These 2-D grids were then smoothed with a 2-D Gaussian kernel ($\sigma$ = 30 ms).

We observed that the classifiers trained around 200 ms post-$S_i$ presentation achieved above-chance accuracy in predicting which $S_i$ category had previously been associated with the presented $S_d$ (the 95th percentile of the peak-level achieved in 200 random shuffle tests is shown as a solid black line in each panel). This effect was above chance from 270–530 ms following presentation of $S_d$. In other words, the spatial pattern of brain activity present 200 ms after presentation of $S_i$ in the Association phase was partially reinstantiated 270–530 ms after presentation of $S_d$ in the Reward phase. Note that the randomization of $S_i$–$S_d$ pairings across subjects makes exceedingly unlikely the possibility that some visual features of $S_i$ happen to be shared with the associated $S_d$ and might therefore carry a shared neural signature.

We also applied the same set of classifiers to the activity evoked by presentation of outcome (coin or neutral blue square) that followed each $S_d$ in the Reward phase. The classifiers trained around 400 ms after $S_i$ achieved above-chance accuracy in predicting the $S_i$ category previously associated with the $S_d$ presented on this trial (*Figure 4B*). This effect was strongest at 70 ms following presentation of the outcome, meaning that the spatial pattern of activity present 400 ms after presentation of $S_i$ in the Association phase was at least partially reinstantiated 70 ms after presentation of the outcome in the Reward phase. Since the outcome always appeared 3500 ms after $S_d$ in each trial, 70 ms after presentation of outcome was equivalently 3570 ms after presentation of $S_d$. Since all the information necessary to retrieve $S_i$ was carried by $S_d$, some of the retrieval process might occur before onset of the outcome.

Two-way ANOVA revealed no significant main effects of 200 ms vs 400 ms or $S_d$ vs outcome but a significant interaction ($p$ = 0.04; *Figure 4C*). That is, the peak accuracy following $S_d$ was higher for the 200 ms than the 400 ms classifier, while the peak accuracy following outcome was higher for the 400 ms than the 200 ms classifier, implying a double dissociation in the component that was more strongly retrieved at $S_d$ vs outcome. Both forms of cross-classification were very much less accurate than (linear) classification of the identity of the $S_d$ (fractals) or outcome (coin/blue square) from the activity directly evoked by these stimuli (cf. *Figure 3C* and *Figure 3—figure supplement 4*).

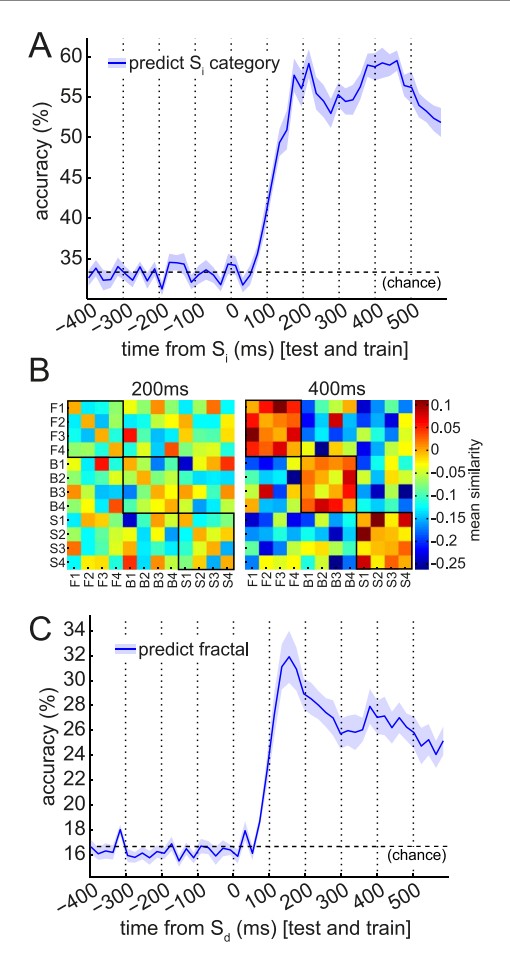

**Figure 3**. Multivariate analysis reveals two temporal components of evoked response to visual stimuli. (**A**) Multivariate decoding performed well to predict the category of photograph ($S_i$) in the Association phase. Cross-validated linear SVM prediction accuracy using all 275 sensors at each time bin is shown. A pattern of two distinct peaks in classifier accuracy around 200 ms and 400 ms after $S_i$ onset is evident. (**B**) At 200 ms after $S_i$ onset, there was no difference in representational similarity between same-category and different-category $S_i$ objects (left panel, p = 0.2 by t-test between subjects). At 400 ms, representational similarity was higher for same-category than different-category objects (right panel, p = $5 \times 10^{-7}$). F1–F4, B1–B4 and S1–S4 refer to the unique faces, bodies and scenes presented during the Association phase. (**C**) When discriminating fractal identity (i.e., a 6-way classification problem of stimuli with no natural categories), performance was sharply peaked before 200 ms after fractal onset. Shaded area shows standard error of the mean.

The following figure supplements are available for figure 3:

**Figure supplement 1**. Univariate classification using best sensor.

*Figure 3. Continued on next page*

To investigate which MEG sensors carried retrieved information, we again trained classifiers on $S_i$-evoked data and tested on $S_d$– or outcome-evoked data (i.e., cross-classification). However, rather than using all 275 sensors, we repeated the procedure for 2000 iterations using a different random subset of 50 sensors each time. To investigate the retrieval identified in *Figure 4A,B*, we restricted analysis to 60 × 60 ms temporal ROIs centered on the peaks of cross-classification in *Figure 4A,B*, and averaged over these temporal ROIs. For each sensor, each iteration of this procedure thus yielded a single classification accuracy. We could then calculate how accurate the cross-classification was on average when a given sensor participated in classification. The average of these data across subjects are shown in *Figure 5*, separately for $S_d$- and outcome-evoked data. To test whether these spatial patterns were significantly different, we again used a linear SVM with cross-validation to predict whether each pattern originated from $S_d$– or outcome-evoked data. Each pattern was mean-subtracted to avoid any trivial classification based on overall higher cross-classification performance for $S_d$- than outcome-evoked data. Prediction accuracy reached 71.2%, which was greater than chance by one-tailed binomial test (p = 0.002).

## Preference for $S_i$+ is correlated with retrieval of stimulus-specific representation at outcome time

Finally, we were intrigued by the apparent retrieval of only the late (400 ms) and not the early (200 ms) component of the $S_i$ representation during outcome presentation. The representational similarity analysis in *Figure 3B* suggested that this 400 ms component might preferentially encode stimulus category. Thus, we speculated the value of the associated $S_i$ category, rather than the value of a particular $S_i$ stimulus, might be updated when the outcome appears. This could provide a potential explanation for the lack of group-level behavioral preference for $S_i$+ over $S_i$– during the subsequent Decision phase, since each $S_i$ category contained both an $S_i$+ and an $S_i$–, with equal presentations. This hypothesis predicts that, although at the group level there might be no significant retrieval of the 200 ms component of $S_i$ representation during outcome presentation, the subjects who did retrieve the 200 ms component of $S_i$ might have a positive preference for $S_i$+ over $S_i$–. (Meanwhile, a preference for $S_i$– over $S_i$+ should be unrelated to retrieval.) We therefore plotted the correlation between behavioral preference and accuracy of $S_i$-trained classifier in predicting the associated category of the $S_d$ stimulus

*Figure 3. Continued*

**Figure supplement 2**. Multivariate classification of $S_i$ for individual subjects.

**Figure supplement 3**. Nearest-mean multivariate classifiers, under a variety of distance metrics, under-perform SVM but extract a similar pattern of multiple peaks in classification performance.

**Figure supplement 4**. Decoding outcome identity.

**Figure supplement 5**. Generalization of instantaneous representational patterns over time, with finer temporal binning.

**Figure supplement 6**. Image statistics.

presented on this trial. This analysis was split according to whether subjects preferred $S_i-$ over $S_i+$ (*Figure 6A*) or $S_i+$ over $S_i-$ (*Figure 6B*). Remarkably, in subjects preferring $S_i+$ over $S_i-$, reinstatement of the 200 ms component of $S_i$ was strongly correlated with behavioral preference. Shuffling subject identities yielded a null distribution of peak $\log_{10}$ p-values for the correlation of classifier accuracy with behavioral preference. The 400 ms classifier showed no substantial positive correlation with behavioral preference (*Figure 6C*), while the 200 ms classifier showed a corrected-significant peak in correlation strength ~400 ms after the onset of the outcome (*Figure 6D*). The raw data driving these correlations are also shown in *Figure 6E,F*.

## Discussion

We used a sensory preconditioning paradigm to explore the temporal structure of the retrieval of representations through associative links. We found that presenting photographs ($S_i$, in three categories) elicited an evolving representation with two temporally distinct components: one around 200 ms and the other around 400 ms after stimulus onset. The earlier component was reinstated when a fractal ($S_d$) previously paired with the $S_i$ was presented. The later component was reinstated when a rewarding or neutral outcome was presented following $S_d$. Although at the group level there was no significant reinstatement of the earlier component at the time of outcome, between subjects the degree of reinstatement of this earlier component correlated with the degree of subsequent value generalization.

Our results fit comfortably with the large body of literature showing that retrieval (which is notably unconscious here and in *Wimmer and Shohamy, 2012*, as contrasted with conscious retrieval that is more commonly studied) induces reinstantiation of at least some aspects of the pattern of neural activity evoked by the original presentation. For instance, in the fMRI study whose design we copied (*Wimmer and Shohamy, 2012*), univariate methods were used to show the equivalent of $S_i$ category retrieval during the Reward phase. Equally, ERP studies have found neural signals as early as 300 ms following a retrieval cue that are different depending on which information is retrieved or whether the information is retrieved (*Johnson et al., 2008*; *Yick and Wilding, 2008*). Further, using MEG, *Jafarpour et al. (2014)* identified reinstatement of a pattern of oscillatory activity appearing approximately 180 ms following presentation of the retrieved item. This pattern was reinstated approximately 500 ms following the retrieval cue, slightly later than the 400 ms we observed.

Multivariate pattern analysis provides a much more powerful microscope than traditional univariate analysis for detecting distributed patterns encoding neural representations (*Norman et al., 2006*). Combining MVPA with MEG enables tracking the fast time-evolution of these representations (*Schmolesky et al., 1998*; *Jafarpour et al., 2013*; *Cichy et al., 2014*). Using these methods we have extended previous findings on retrieval to now establish a mapping between the dynamics of object representation and the dynamics of retrieval in this behavioral paradigm.

We identified two temporal components of object representation that were retrieved at different times. The earlier component of $S_i$ representation, which appeared roughly 200 ms following $S_i$ presentation, was first detectable 270 ms following presentation of $S_d$. This is consistent with past ERP studies showing similar timing, which have been taken as suggesting that reactivation is mediated by hippocampus (*Bosch et al., 2014*). The prolongation of this representation from 270–530 ms may represent averaging (over trials or subjects) of temporally abrupt retrievals, or a sustained information retrieval.

By contrast, the late component of $S_i$ representation re-appeared 70 ms following outcome presentation. The outcome did not provide any additional information about $S_i$ category, so the representation of $S_i$ must have been sustained in some form through the (fixed) delay between $S_d$ and outcome. This raises questions such as where the information about $S_i$ was held during the delay, and what are the implications of this timing. For the former, we were only able to detect a representation of $S_i$ when it took the form of a spatial pattern of activity mirroring the pattern at presentation of $S_i$. Thus

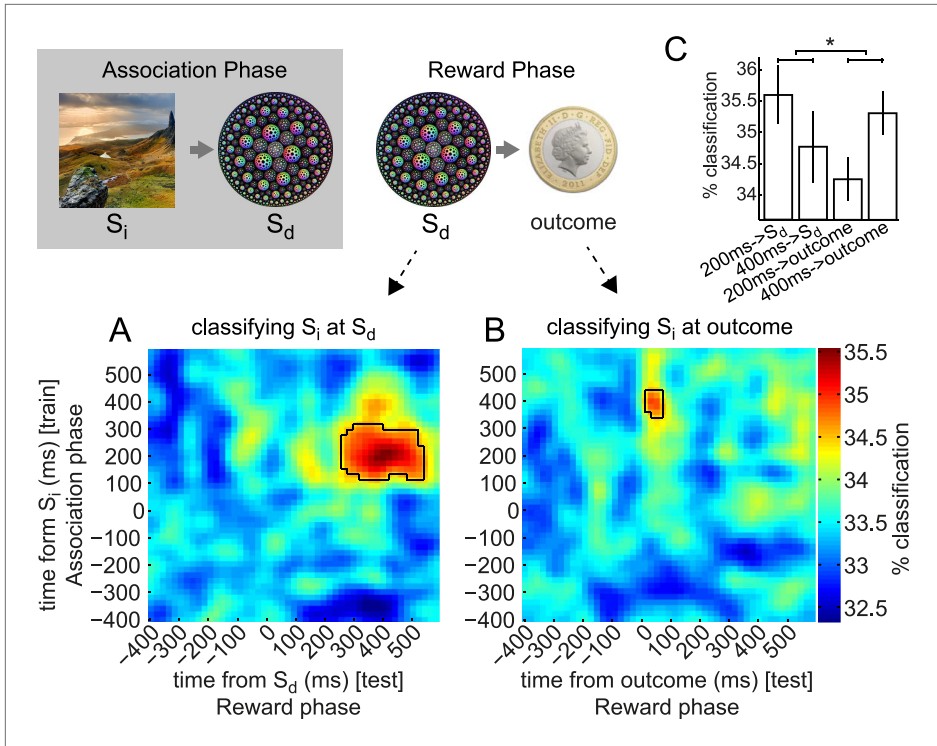

**Figure 4**. Early and late components of associated object representation retrieved at time of cue and outcome, respectively. During the Reward phase, the 200 ms component of the $S_i$ representation was retrieved for an extended period from shortly after $S_d$ was presented, while the 400 ms component of $S_i$ representation was retrieved around the time the outcome was presented. (**A**) Classifiers trained around 200 ms after $S_i$ presentation in Association phase and tested around 400 ms after $S_d$ presentation in Reward phase decode the object category previously associated with the $S_d$. Photo is from pixabay.com and is in the public domain. (**B**) Classifiers trained around 400 ms after $S_i$ presentation and tested 70 ms after outcome presentation decode the object category previously associated with the $S_d$. In **A** and **B**, black outlines show p = 0.05 peak-level significance thresholds (empirical null distribution generated by 1000 random permutations of training category labels, see Methods for more details). (**C**) Peak classification accuracy in the 200 ms and 400 ms rows of A and B. By 2-way ANOVA, there was no main effect of 200 ms vs 400 ms or of $S_d$ vs outcome, but there was a significant interaction (p = 0.04). Error bars show standard error of the mean.

information might have been online in the activity of, for instance, prefrontal neurons (*Fuster, 2001*; *Wang et al., 2006*), but in a different form from that inspired by $S_i$ itself (*Sakai and Miyashita, 1991*; *Rainer et al., 1999*). Alternatively, it might have been stored in short-term synaptic weight changes (*Hempel et al., 2000*; *Seung, 2003*; *Florian, 2007*; *Mongillo et al., 2008*).

Supporting the idea of these ~200 ms and ~400 ms components as distinct representational periods, we note the following. First, there was a decrease in classification accuracy between these periods. Second, classifiers trained on one epoch had low accuracy in the other epoch (*Figure 3—figure supplement 5*), suggesting information about the stimuli was coded differently between epochs. Third, the epochs had different similarity structure with respect to the stimulus categories (*Figure 3B*). Fourth, the patterns from the two epochs were doubly dissociated in terms of their retrieval at $S_d$ vs outcome (*Figure 4*), while the time period between the two peaks (i.e., around 300 ms post-stimulus) was not strongly retrieved either at $S_d$ or outcome (*Figure 4*).

In terms of timing, the relatively precise epoch of retrieval of $S_i$ following the presentation of the outcome may reflect the point of strongest overlap between a variety of timings in individual subjects. Alternatively, it may be that a representation that is latent became detectable as soon as more power arose in the visual-evoked ERF due to onset of the outcome. Yet another possibility is expectations of the next stimulus partly drive representations in the first 10 s of milliseconds after a visual onset, before the present stimulus is processed.

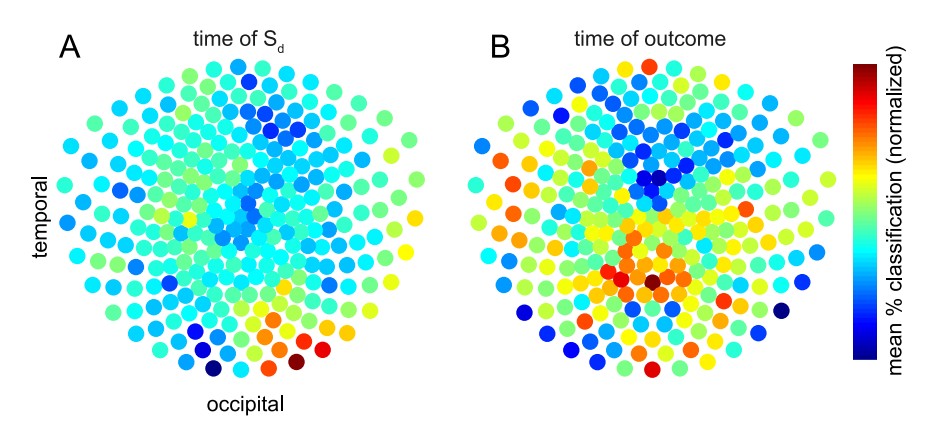

**Figure 5**. Contributions of sensors to retrieval. To explore which brain areas carried the information about $S_i$ that was retrieved at the time of $S_d$ and outcome, we copied the procedure of training linear category classifiers on presentation of $S_i$, and predicting the category at the time of $S_d$ or outcome—but instead of using all 275 sensors, we repeated the analysis 2000 times using subsets of 50 sensors randomly selected on each iteration. The contribution of sensor $s$ was taken to be the mean of all prediction accuracies (within 60 × 60 ms temporal ROIs containing the peak time bins) achieved using an ensemble of 50 sensors that included $s$. Intriguingly, the information about the category of $S_i$ retrieved at the time $S_d$ was presented emerged primarily from occipital sensors (**A**), while the information about the category of $S_i$ retrieved at the time the outcome was shown appeared more strongly in parietal and temporal sensors (**B**). In the difference between the two conditions, no individual sensor survived correction for multiple comparisons. However, a linear SVM was reliably able to classify whether a spatial pattern belonged to $S_d$ or outcome (71.2% accuracy, p = 0.002 by one-sided binomial test against chance classification).

The low accuracy in classifying retrieved representations (~35%) compared to evoked responses (~60%) might imply that retrieved representations (perhaps especially those that subjects are not consciously aware of) were weak compared to evoked representations. It is also possible that $S_i$ representations were only retrieved on a subset of trials, weakening the average signal. Finally, it is possible that retrieved representations had a distributed spatial pattern that was only partly overlapping with the evoked representation, making it more difficult to detect with pattern classifiers trained on evoked activity.

We exploited the distinct temporal components of retrieval to help elucidate the neural underpinnings of value generalization through associations. In both our study and in the similar design of *Wimmer and Shohamy (2012)*, behavioral evidence of sensory preconditioning rests wholly on stimulus-specific retrieval (since the rewards associated with each category are balanced). If the 400 ms component of $S_i$ representation preferentially encodes information about category rather than specific stimuli, as suggested by our representational similarity analysis, retrieval of solely this component at outcome time might cause value learning to be assigned to categories rather than individual stimuli. This hypothesis would explain our finding that the subjects who retrieve the 200 ms component at outcome show behavioral evidence of sensory preconditioning. Under this interpretation, the correlation that Wimmer and Shohamy found in BOLD between retrieved stimulus representations and behavior between subjects may also have been driven by the 200 ms component of the stimulus representation; these temporally precise signals could not be distinguished using fMRI. Although the particular representations online at the time of reward were probably driven by quirks of this task design (since other sensory preconditioning experiments have found robust group-level preference for $S_i$ paired with rewarded $S_d$ (e.g., *Seidel, 1959*)), the finding is of general importance because it suggests that the exact timing of reward relative to fast-evolving neural representational structures is crucial to value updating and credit assignment.

Like *Wimmer and Shohamy (2012)*, we have compared a behavioral value generalization measure against the output of a neural classifier trained on the category of $S_i$, rather than the identity of an individual $S_i$. The latter would give a more direct test of the idea that subjects who retrieve a representation of the specific $S_i$ paired with the particular $S_d$ viewed on this trial drive larger value updates.

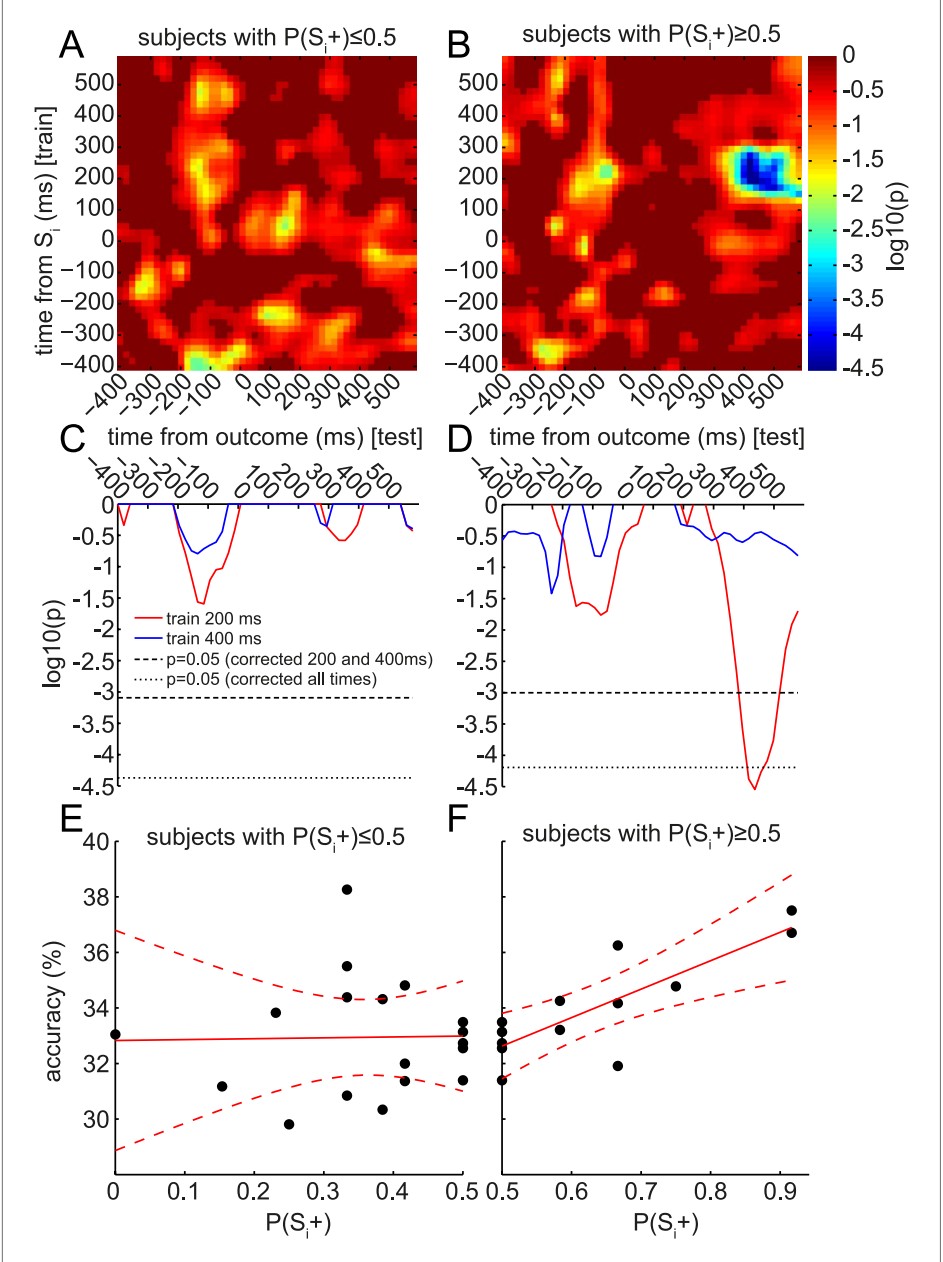

**Figure 6**. Retrieval of early component of $S_i$ representation predicts value updating across subjects. At the group level, only the 400 ms component was significantly retrieved at the time of outcome (cf. **Figure 4B**). However, at the single-subject level, the degree of retrieval of the 200 ms component correlated with value updating. As in **Figure 4B**, the accuracy of classifiers trained at each time bin around $S_i$ (in the Association phase) was tested at each time bin around the time of outcome (in the Reward phase) to predict the category of the $S_i$ associated with the $S_d$ preceding the outcome. In each time*time bin, this accuracy was regressed, across subjects, against the behavioral preference for $S_i+$ over $S_i-$ from the Decision phase (i.e., $P(S_i+)$). As we only explored positive correlations, one-tailed $log_{10}$ p-values of the regression are reported. (**A**) In subjects who preferred $S_i-$ over $S_i+$, there were no correlations between the degree of preference and the degree of reinstatement of $S_i$ at outcome. (**B**) In subjects who preferred $S_i+$ over $S_i-$, there was a strong correlation between the degree of preference and the degree of reinstatement. This correlation peaked at around 400 ms after outcome onset. (**C**, **D**) Red and blue traces show single rows of panels A and B at 200 and 400 ms. Significance was tested by randomly shuffling subject identities to obtain a null distribution of peak-level log10 p-values. Thresholds are shown at 95% of the null distribution of the peak-level of 200 and 400 ms rows, and at 95% of the null distribution of peak-level of all rows. (**E**, **F**) Raw classification accuracies underlying the correlations in **A**–**D**, when training at 200 ms after $S_i$ onset and testing at 400 ms after outcome onset. Each point is a subject.

Although it is in principle possible to train a classifier to distinguish between individual exemplars of an $S_i$ category, this did not reach a sufficiently high level of performance in our hands, perhaps limited by the relatively small number of training samples per unique stimulus. Future experiments could also employ $S_i+$ and $S_i-$ stimuli that are more neurally distinguishable.

We noted in the 'Introduction' a large number of proposals for the use of associative information both at the time of decision (online) or when a decision is not imminent (offline). Offline and online processes may share similar mechanisms (*Doll et al., 2014*), and in some cases the division between offline and online mechanisms is blurred. For example, retrieving elements of past experiences may serve as part of the process of planning in advance for the next time related situations are encountered (*Dragoi and Tonegawa, 2011*, *2013*), similar to the psychological notion of implementation intentions (*Gollwitzer, 1999*).

Some theoretical methods (e.g., the successor representation (*Dayan, 1993*) and beta-models [*Sutton, 1995*]) shift a portion of the burden of online calculations using offline updates to carefully structured representations. In sensory preconditioning, it is an open question whether generalized values are updated offline (either during the Reward phase or in between the Reward and Decision phases), retrieved through associative links at the time of decision, or a mix of both. In animals the vulnerability of sensory preconditioning to extinction (*Gewirtz and Davis, 2000*) hints at an online mechanism, but it is equally possible that extinction drives offline value updates through the same generalization mechanism as acquisition. Indeed, although our description of the reinstatement of $S_i$ suggests that it arises through a distinct process of retrieval, we cannot distinguish this from the subtly different possibility hinted by these ideas that the representation of $S_d$ changed through the associative learning so that it more closely resembles that of $S_i$.

In animals, the temporal structure of retrieval appears to subserve complex memory (*Sirota et al., 2003*; *Schwindel and McNaughton, 2011*), learning and decision-making processes, especially in hippocampus and hippocampal–cortical interactions. Rodents retrieve representations of past and future locations, actions, and rewards (*Johnson and Redish, 2007*; *van der Meer et al., 2010*; *Steiner and Redish, 2014*); the timing of this retrieval is tightly structured and likely encodes critical information in the decision-making computation. In humans, frontal theta power (*Hsieh et al., 2011*) and patterns of activity in hippocampus (*Ezzyat and Davachi, 2014*; *Hsieh et al., 2014*) are implicated in coding temporal order within sequences of stimuli. Applying methods from the present work could be useful to establish a finer grained map of the representations used in complex memory and decision processes.

Important to understanding the retrieval dynamics in this behavioral paradigm is the shift we observed in the dominant coding of information in evoked responses from 200 ms to 400 ms post-stimulus. Information in the visual system up to 200 ms post-stimulus may hew closely to the form of the stimulus that was presented (*Tanaka and Curran, 2001*; *VanRullen and Thorpe, 2001*; *Liu et al., 2002*; *Schiff et al., 2006*; *Rossion and Jacques, 2008*). This is consistent with our finding that spatial patterns of activity evoked by different exemplars within a category were relatively distinct and that individual fractals were better classified at this time bin. Conversely, brain activity later than 200 ms post-stimulus is often found to include contextual and other sources of information (*Kok, 2001*; *Tsivilis et al., 2001*; *Schiff et al., 2006*; *Garrido et al., 2007*; *Sanguinetti et al., 2014*). In particular, the N400 component of the event-related potential (ERP) in EEG extends from roughly 250–500 ms post-stimulus and appears to be driven at least partly by the medial temporal lobe, which is functionally coupled to sensory cortices (*Bar, 2004*). The N2pc component of the ERP, which occurs earlier from roughly 200–300 ms post-stimulus, has also been tied to contextually-sensitive processing (*Conci et al., 2006*; *Schiff et al., 2006*), and originates from lateral temporal and parietal sources (*Hopf et al., 2000*; *Oostenveld et al., 2001*). Although information about the category of our stimuli is directly available in their visual form, one interpretation of our observation of more consistent category information at 400 ms is that this reflects such contextually-sensitive processing happening based on lateral and top-down functional connections (*MacKay and Bowman, 1969*; *Rao and Ballard, 1999*; *Ullman, 2000*; *Engel et al., 2001*; *Bledowski et al., 2006*; *Garrido et al., 2007*; *Friston and Kiebel, 2009*; *Kourtzi and Connor, 2011*).

Finally, we note that timing of event-related signals depends strongly on stimulus properties (e.g., *Bobak et al., 1987*). Multivariate classification also yields different timings in the peaks of classification depending on the specific kinds of categories evaluated (*Cichy et al., 2014*). Thus the particular temporal structure of evoked responses is most likely specific to the stimuli used. Mapping this structure for a given task and stimuli can be leveraged to probe the dynamics of retrieval.

In summary, neural retrieval of representations through associative links is central for memory and decision-making. Here we provide evidence that the dynamical structure within retrieval is functionally relevant for value-guided decision making. Analyzing the fine temporal structure of representations also increases the potential for studying temporally rich retrieval processes such as replay and planning in humans, which were previously confined to animal recordings.

## Materials and methods

### Subjects

Twenty-nine adults participated in the experiment, recruited from the UCL Institute of Cognitive Neuroscience subject pool. Three were excluded before the start of analysis for large movement or myographic artifacts. Of the 26 remaining, age quartiles were 18.7, 19.5, 21.3, 26.7, 41.4 years; 14 were female, and 1 was left-handed. All participants had normal or corrected-to-normal vision and had no history of psychiatric or neurological disorders. All participants provided written informed consent and consent to publish prior to start of the experiment, which was approved by the Research Ethics Committee at University College London (UK), under ethics number 1825/005.

### Task

Participants performed three phases of a simple behavioral task (copied almost exactly from **Wimmer and Shohamy, 2012**; but with timings set to be faster for MEG) designed to induce and measure sensory preconditioning. The task was coded in Cogent (Wellcome Trust Centre for Neuroimaging, United Kingdom), running in MATLAB 7.14 (Mathworks, Natick, Massachusetts).

Before the experiment, participants rated 78 images, one at a time, using a visual analog scale to indicate how much they subjectively liked each image, ranging from 'Strongly Dislike' to 'Strongly Like'. These images consisted of 60 photos (20 faces, 20 body parts, 20 scenes), and 18 fractals. Luminance and contrast varied between images (**Figure 3—figure supplement 6**). Four of each photo category and 12 fractals were then selected to be used in the experiment. For each subject we chose the stimuli whose liking ratings were closest to neutral; different subjects therefore saw different images in the experiment.

In the first ('Association') phase of the experiment, each of the 12 selected photos ('$S_i$', indirect stimuli) were deterministically paired with a different fractal pattern ('$S_d$', direct stimuli). Two of each $S_i$ category were 'dummies' for the cover task, and two were 'real' stimuli. Subjects viewed $S_i$ and $S_d$ images sequentially while performing a cover task of pressing one button in response to rightside-up images and a different button for upside-down images, with the button response mapping randomized across subjects. Dummies had a 50% chance of being upside-down, and real stimuli were never upside-down. Dummies were not presented in subsequent phases. In each trial, subjects saw an $S_i$ for 1750 ms, followed by an interstimulus-interval (ISI) of 1000 ms, followed by the paired $S_d$ for 1750 ms, followed by an intertrial-interval (ITI) of 2500 ms. Every nine trials, each of the six real $S_i$ stimuli was presented once, and one of each of the dummy $S_i$ stimuli in each category was presented once (both reals and dummies were always followed by the paired $S_d$). The order was randomly permuted over every 9 trials, and this was repeated 12 times, for a total of 108 trials. In debriefing at the end of the experiment, no subject reported being aware of any pairing between $S_i$ and $S_d$ indicating the effectiveness of the cover task; the $S_i$–$S_d$ association was implicit. No subject reported being aware that the dummies did not appear in later phases.

In the second ('Reward') phase, subjects were taught that some of the fractals ($S_d$+) were worth money, while others ($S_d$−) were not. In each conditioning trial, subjects saw an $S_d$ for 2000 ms, followed by an ISI of 1500 ms, and then either a reward (image of a one pound sterling coin) or no-reward (blue square) for 2000 ms, followed by an ITI of 3000 ms. Each $S_d$ appeared 18 times, for a total of 108 trials. $S_d$− were never rewarded, while $S_d$+ were rewarded 14 out of 18 times that they appeared. The cover task was to press one button for any $S_d$ or for no-reward, and a different button for reward (meaning that in an unrewarded trial, the same button was to be pressed twice; while in a rewarded trial two different buttons should be pressed). Pressing the correct button to 'pick up' the coin led to actually receiving this money at the end of the experiment (divided by a constant factor of ten); subjects were informed of this. Through the unique pairing between $S_i$ and $S_d$, conditioning implicitly established $S_i$+ (previously paired with $S_d$+) and $S_i$− (previously paired with $S_d$−). The pairing was such that each $S_i$ category contained one $S_i$+ and one $S_i$−.

In the third ('Decision') phase, in each trial subjects made a pairwise choice between either two $S_d$ images or two $S_i$ images. The two $S_i$ images were always of the same category (face/body/scene): one $S_i+$ and one $S_i-$; likewise, the two $S_d$ images, an $S_d+$ and an $S_d-$, had always been previously paired with the same $S_i$ category. Subjects were instructed that they would receive monetary reward for choosing the correct stimulus, but, as in *Wimmer and Shohamy (2012)*, were given no instructions about how to identify the correct stimulus (except to choose the one they thought was more lucky). They actually received these rewards at the end of the experiment, again divided by ten. In addition to the money earned within the task, subjects received a flat compensation of £10. Each pairwise choice was repeated 4 times for a total of 24 trials. Any preference for $S_i+$ over $S_i-$ would provide evidence of sensory preconditioning.

After the experiment, subjects again provided subjective liking ratings on a visual analog scale, this time for each $S_i$ and $S_d$ actually used in the experiment (excluding dummies).

## Behavioral analysis

Decision-phase preferences for $S_d+$, $S_d-$, $S_i+$, and $S_i-$ were measured by averaging the four binary responses for each pair, and performing a one-sample t-test between subjects on the mean response against 50%. Similar results could be obtained by treating the first choice of each subject for each pair as an independent draw from a Bernoulli distribution and comparing the results to p = 0.5. Changes in subjective liking ratings from Pre-Liking to Post-Liking phases were differences on an arbitrary scale (pixels in the visual analog scale) and were linearly de-trended as subjects showed a robust tendency to increase all ratings at the end of the experiment compared to the beginning (many subjects reported in debriefing that they liked most of the stimuli more because they were more familiar at the end of the experiment).

## MEG acquisition

MEG was recorded continuously at 600 samples/second using a whole-head 275-channel axial gradiometer system (CTF Omega, VSM MedTech, Canada), while participants sat upright inside the scanner. Continuous head localization was recorded with three fiducial coils at the nasion, left pre-auricular, and right pre-auricular points. The task script sent synchronizing triggers (outportb in Cogent) which were written to the MEG data file. A projector displayed the task on a screen ~80 cm in front of the participant. Participants made responses on a button box using either thumbs or index fingers as they found most comfortable.

## MEG analysis

All analysis was performed in MATLAB. Some analyses used SPM12b (Wellcome Trust Centre for Neuroimaging, United Kingdom). Data were first converted to SPM12 format using spm_eeg_convert. Each event was then epoched, using spm_eeg_epochs, to 1000 ms segments from −400 ms to +600 ms relative to the event, based on the triggers recorded from the task script. All timings were corrected for one frame (1/60 s) of lag between triggers and refreshing of the projected image, measured using a photodiode outside the task. The 600 samples in each epoch were then reduced to 50 time bins by averaging together each consecutive 12 samples. Thus, the time bins were spaced every 20 ms and represented the average raw signal of the 12 samples within that 20 ms. Pre-stimulus bins were treated as baseline.

We built three-way classifiers for the category of the $S_i$ stimuli. Classifiers were trained based on the activity evoked by the presentation of the $S_i$ stimuli in the Association phase, and used to classify the activity associated with the presentation of the $S_d$ and outcome stimuli in the Reward phase. Classifiers were built for each time bin following $S_i$ presentation, and tested on each time bin following $S_d$ and outcome presentation during the Reward phase, giving rise to (Association) time*(Reward) time maps of classification performance.

Support vector machine (SVM) classification analyses were performed with the svmtrain/svmpredict routines from libsvm (National Taiwan University, Taiwan; http://www.csie.ntu.edu.tw/~cjlin/libsvm). Each feature used for classification (i.e., a sensor at a time bin) was independently z-transformed before classification. Results are reported with linear kernels. The regularization parameter C was tuned to optimize cross-validation performance in cross-validation of Association-phase data (C = $10^5$) but was then fixed for all further analyses. Cross-validation was tested using leave-one-out, k-fold (5, 10, or 20), or repeated random subsampling (50 or 100 independent subsamples with 10% of samples left-out), without any difference in results between methods.

In *Figure 4*, we show 2-dimensional maps where the dimensions are times relative to two different events. To generate statistical significance thresholds for these maps, we recalculated these maps many times with independently shuffled category labels for the stimuli. Each shuffle yielded a map that contained no true information about the stimuli, but preserved overall smoothness and other statistical properties. The peak levels of each of these maps were extracted, and the distribution of these peak levels formed a nonparametric empirical null distribution. The 95th percentile of this distribution is reported as the significance threshold.

Representational similarity between two different trials was measured by correlation between the patterns of activation over sensors, at the same time bin relative to stimulus onset.

Classifiers trained on Association-phase data were used directly to predict Reward-phase data without any tuning to optimize cross-classification performance. All (Association) time*(Reward) time maps of classification performance were smoothed by a 2-D Gaussian kernel ($\sigma$ = 30 ms) for display and for calculating peak-level shuffling statistics.

### Analyses that didn't work

In the interest of reporting our work as completely as possible, we discuss a set of analyses that were based on relevant hypotheses, but did not lead to significant results.

1) An important issue in the analysis of retrieved representations is to make sure that what are apparently retrieved representations are not in fact coincidences in the representation of the retrieved object and the retrieval cue. In the analyses in the main paper, this is controlled by randomizing $S_i$–$S_d$ pairings between subjects. We attempted another way of controlling for this, by training a classifier on all subjects' (except one) $S_d$-evoked data (using the category labels of the associated $S_i$s), and testing on the left-out subject. If this procedure, repeated across left-out subjects, would produce an above-chance prediction of the $S_i$ category associated with the displayed $S_d$, this would imply that the $S_d$-evoked data contain a real representation of $S_i$. Unfortunately when we attempted this, the group-level prediction of $S_i$ category did not reach significance. We speculate this is because the category representation differs substantially between subjects (supported by *Sandberg et al. (2013)*); an issue that the analysis in the main paper is immune to because classifiers are trained separately for each subject.

2) *Wimmer and Shohamy (2012)* regressed their neural signal against within-category differences in behavioral preference. For example, if one subject in the Decision phase preferred the face paired with the rewarded fractal, but did not prefer the scene paired with the rewarded fractal, then he or she was more likely to have a large fusiform face area activation during presentation of the face-paired fractal in the Reward phase than to have a large parahippocampal place area activation during presentation of the scene-paired fractal. We attempted the same analysis but no correlation with neural decoding reached significance. In our hands collapsing within categories to look at between-subject variance in total value updating appeared more statistically powerful. Along similar lines, we also trained classifiers to distinguish individual stimuli in the Association phase (e.g., a particular face, rather than the category of faces—so the classifier learned about 12 distinct categories), and applied these classifiers to activity at the time of outcome in the Reward phase. The classifier was treated as 'correct' if it predicted the identity of the photograph that had been previously associated with the fractal presented on this trial of the Reward phase. We then correlated the resulting correctness ratings against the behavioral preference for $S_i+$ over $S_i-$ in the Decision phase (just as in *Figure 6* of the main paper, but classifying individual stimuli rather than categories). However, these correlations did not reach shuffle-corrected significance. This may be a result of the difficulty of classifying many individual stimuli with relatively few trials.

3) We wondered if, when photos ($S_i$) were presented during the Decision phase, it would be possible to identify neural signals containing information about the paired fractal ($S_d$). It is possible that this could represent an online retrieval of value information about $S_d$ to guide the choice about $S_i$. However, we could not detect above-chance classification of either associated $S_d$ when pairs of $S_i$ were presented during the Decision phase. We suspect the patterns of representation may be more difficult to disentangle when two stimuli are shown on-screen at the same time.

## Acknowledgements

We thank Anna Jafarpour, Will Penny, Aidan Horner, Martin Hebart and Tim Behrens for helpful discussions and Laurence Hunt and 'Ōiwi Parker Jones for comments on an earlier version of this manuscript.

## Additional information

### Funding

| Funder | Grant reference number | Author |
|---|---|---|
| Wellcome Trust | 091593/Z/10/Z | Zeb Kurth-Nelson, Gareth Barnes, Ray Dolan |
| Max-Planck-Gesellschaft | UCL Centre for Computational Psychiatry and Ageing Research | Zeb Kurth-Nelson, Ray Dolan |
| Gatsby Charitable Foundation | | Dino Sejdinovic, Peter Dayan |
| Medical Research Council | MR/K005464/1 | Gareth Barnes |
| Wellcome Trust | 098362/Z/12/Z | Ray Dolan |

The funders had no role in study design, data collection and interpretation, or the decision to submit the work for publication.

### Author contributions

ZK-N, Conception and design, Acquisition of data, Analysis and interpretation of data, Drafting or revising the article; GB, PD, Conception and design, Analysis and interpretation of data, Drafting or revising the article; DS, Analysis and interpretation of data, Drafting or revising the article; RD, Conception and design, Drafting or revising the article

### Ethics

Human subjects: All participants provided written informed consent and consent to publish prior to start of the experiment, which was approved by the Research Ethics Committee at University College London (UK), under ethics number 1825/005.

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
