## [Decision Letter]

Thank you for sending your work entitled “Temporal structure in the retrieval of associations” for consideration at *eLife*. Your article has been favorably evaluated by a Senior editor, a Reviewing editor, and 3 reviewers.

The following individuals responsible for the peer review of your submission have agreed to reveal their identity: Howard Eichenbaum (Reviewing editor), Nikolaus Kriegeskorte (peer reviewer), and Kenneth Norman (peer reviewer). A further reviewer remains anonymous.

The Reviewing editor and the reviewers discussed their comments before we reached this decision, and the Reviewing editor has assembled the following comments to help you prepare a revised submission.

The reviewers were consensual in their enthusiasm for the question under study and for the sophisticated design and analyses, for the novel findings on temporal dynamics of memory reactivation, and the inclusion of negative findings. There were, however, major concerns about the strength of some of the findings, and there were other specific concerns by individual reviewers about some of the statistical tests, the exposition, and the interpretation of specific findings. These concerns are provided below.

Reviewer 1:

Reinstatement is only significant at the level of category-average representations, not individual exemplars of each face, body, and scene. This is a pity because each category was equally often associated with positive and neutral reward. This makes it difficult to assess whether the specific reward-predictive stimuli were reinstated at the point of reward and whether the strength of such reinstatement predicted the propagation of value updates. The authors speculate that category-level reinstatement might explain the absence of significant value updating suggested by their behavioural post-test. They do show that stronger reinstatement of the early component of the category representation just before reward onset predicts subjects' value-based choices in the behavioural post-test. However, although reasonably corrected for multiple testing across latencies, this effect is barely significant and is one of several analyses attempting to establish a relationship between reinstatement of reward-predictive stimuli and value updating. [Reviewing editor: This comment clarifies the reviewers' concern about this issue, but it may not be possible to improve on the comments provided by the authors in the discussion of their findings.]

Reviewer 2:

1) The Introduction does a nice job of identifying a gap in the literature, though I believe it overstates this gap in some ways. I would suggest additional discussion of the findings of Manning et al. (2011; 2013, Journal of Neuroscience), which identify the spatial and spectral signatures of reinstatement. Perhaps the authors could emphasize the temporal aspects of associative retrieval more thoroughly in this section.

2) The authors should more clearly identify the cognitive constructs they are assessing. Throughout the manuscript, the authors seem to alternate between focusing on a paradigm designed for associative retrieval, sensory preconditioning, and value updating. Given that the task design allows for investigating each of these, the authors should state this upfront and additionally discuss the prior literature linking these constructs in the Introduction.

3) The rationale for using univariate vs. multivariate feature sets should be stated earlier in the text. Is the rationale only to verify that more information is captured using multivariate techniques compared to univariate techniques? If so, I think simply citing the prior literature demonstrating this should be sufficient and it is unnecessary and a bit tangential to focus on this point as a primary finding. As the text stands now, it takes a while to get to the point of the primary research findings.

4) Pattern classification performance was calculated within subject and then averaged across subjects, revealing peaks around 200ms and 400ms. To what extent are these peaks evident in individual subjects? The authors partially raise this issue themselves in the Discussion section, but may wish to provide quantitative results regarding this issue.

5) Although the data generally supports the claim that they are primarily measuring “evoked” activity, the authors should be cautious to not imply that it is only evoked activity.

6) In the Discussion section: Several EEG and fMRI studies of temporal order memory (some using an RSA approach) are excluded but should be cited and/or discussed (Hsieh, 2011, Journal of Neuroscience; [32], Neuron; [17], Neuron).

7) Overall, the data support the claim of two temporally distinct representational periods. However, such an interpretation is complicated by a highly significant (double the chance rate) classification performance during the two windows. The authors should discuss this more thoroughly. Additional analyses described above may also clarify this interpretational discrepancy.

Reviewer 3:

1) What kinds of multiple comparisons corrections are the authors using for the analyses shown in Figure 4? Elsewhere in the paper, the investigators appear to be using an approach of controlling for familywise error at p < .05, but it is not clear what is going on here (the paper only says p = 0.05 “peak-level significance thresholds”). Whatever correction the authors use will need to correct for the use of classifiers trained on different time points and also their exploration of multiple time points during the reward learning phase. For what it's worth, I thought that the approach taken in Figure 6 (controlling for all time points, but only two classifiers: the 200ms-trained classifier and the 400ms-trained classifier) was acceptable; a similar approach could be applied here, if it isn't being used already.

2) For the RSA analysis, it was not clear if the authors obtained a meaningful measure of “self-similarity” values (i.e., how similar is the pattern for a particular item to the pattern evoked by other instances of that same item). If they did not measure this, it would be useful to re-do the analysis in a way that obtains self-similarity measurements. Having this information would allow the authors to get separate readouts of item-specific information (e.g., by contrasting the diagonal cells to the off-diagonal cells from the same category) and category information. In particular, it would be useful to get these measures (along with some statistical assessment of their reliability, e.g., through bootstrapping) for both the 200ms and 400ms time points. While it is clear that category structure is not strongly represented at the 200ms time point, the RSA analysis in its current form does not speak to how strongly item-specific information is represented at the two time points (the text, as it is currently written, seems to be implying that there is less information about individual items at 400ms than 200ms, but we don't know that).

3) As shown in Figure 1, about half of the participants showed P(S_i_+) that was below .5, and some of these participants showed P(S_i_+) values that were well below .5. While I can understand how variance in the level of reactivation (during the reward learning phase) could lead to variance in participants' preference, ranging from no preference to strong preference in favor of the associated item, but it is not clear how variance in reactivation could lead to the opposite preference. The only other interpretation of the below-chance performance is that it is noise, but in that case, how can it be explained using classifier evidence? It would be useful if the authors commented on this.

4) A suggestion: to the extent that classification performance is suffering due to a lack of training data, we have sometimes found in my lab that we can improve classification by using a “leave one subject out” approach (i.e., train on all but one subject, test on the left-out subject). This approach assumes that brain patterns are relatively consistent across participants. If that assumption is generally true, then the 25-fold increase in the number of training patterns can improve classification accuracy by a substantial margin (conversely: if there is extensive between-subject variability in the patterns, then moving to a leave-one-subject out approach can hurt classification accuracy). Anything that improves classification accuracy has the potential to greatly boost the interpretability of these results.

---

## [Author Response]

We thank all the reviewers for their detailed and insightful comments and questions, which greatly improved the clarity of the paper, as well as uncovering a new analysis that we believe strengthens its final result.

Reviewer 1:

*Reinstatement is only significant at the level of category-average representations, not individual exemplars of each face, body, and scene. This is a pity because each category was equally often associated with positive and neutral reward. This makes it difficult to assess whether the specific reward-predictive stimuli were reinstated at the point of reward and whether the strength of such reinstatement predicted the propagation of value updates*.

Thanks very much for the comments. We agree. It would be great to see convincing reinstatement at the level of individual exemplars. We suspect getting the power to resolve this would require more presentations of each individual stimulus.

*The authors speculate that category-level reinstatement might explain the absence of significant value updating suggested by their behavioural post-test. They do show that stronger reinstatement of the early component of the category representation just before reward onset predicts subjects' value-based choices in the behavioural post-test. However, although reasonably corrected for multiple testing across latencies, this effect is barely significant and is one of several analyses attempting to establish a relationship between reinstatement of reward-predictive stimuli and value updating. [Reviewing editor: This comment clarifies the reviewers' concern about this issue, but it may not be possible to improve on the comments provided by the authors in the discussion of their findings*.*]*

Yes, we completely agree. We added the “failed analyses” section and the more conservatively corrected threshold in Figure 6 to try to be upfront about this. Originally we were reasonably happy that the effect did survive modest correction, even with subjects being unconscious of the associations.

But, in response to Reviewer 3, we split the correlation between decoding and behavior according to positive (preferring S_i_+ over S_i_-) or negative (preferring S_i_- over S_i_+) preference. We now find that looking only at people with a positive behavioral effect greatly improves the strength of the correlation (new Figure 6).

This finding is consistent with Reviewer 3's observation that above-chance decoding could be linked to a positive behavioral effect, but below-chance decoding should not be linked to a negative behavioral effect. We thank the reviewer for this insight and this now strengthens the significance of the relationship between reinstatement and value updating.

Reviewer 2:

*1) The Introduction does a nice job of identifying a gap in the literature, though I believe it overstates this gap in some ways. I would suggest additional discussion of the findings of Manning et al. (2011, 2013, Journal of Neuroscience), which identify the spatial and spectral signatures of reinstatement. Perhaps the authors could emphasize the temporal aspects of associative retrieval more thoroughly in this section*.

Thank you for the suggestion. We have added the Manning citations (2013, JoN, should be 2012, JoN, right?), mentioning explicitly how they investigated oscillatory patterns. We have also tried to frame more clearly the specific gap and the question we're looking at (i.e., decomposing the trajectory of spatial patterns that appear at study and seeing which of these elements reappear at retrieval).

This part of the Introduction now reads:

“… electrophysiology experiments robustly demonstrate that when an object is directly experienced, the evoked pattern of neural activity evolves rapidly over tens to hundreds of milliseconds (8; 51; 59; 70; 76, 76; 77; 93). This implies that direct experience of an object evokes multiple distinct spatial patterns of neural activity in sequence. However, these distinct spatial patterns have never been identified independently at retrieval.

At retrieval, recent studies have explored the fast evolution of neural representation. EEG studies provide evidence that some information is retrieved as early as 300ms following a cue (e.g., [39]; [95]; [97]). [52], [53], using electrocorticography, and [36], using MEG, showed that oscillatory patterns are also reinstated during retrieval; in two of these studies the predominance of low oscillatory frequencies in reinstatement suggests a potential spectral signature.

However, the dynamics of representation during direct experience of an object have never been tied to the dynamics of retrieval. It is not known which of the patterns evoked in sequence by direct experience are reinstantiated during retrieval, what the temporal relationship is in their retrieval, or what functional significance this has.”

*2) The authors should more clearly identify the cognitive constructs they are assessing. Throughout the manuscript, the authors seem to alternate between focusing on a paradigm designed for associative retrieval, sensory preconditioning, and value updating. Given that the task design allows for investigating each of these, the authors should state this upfront and additionally discuss the prior literature linking these constructs in the Introduction*.

Thank you for this suggestion. We are now more explicit in the Introduction, by first introducing the sensory preconditioning paradigm, and then describing how associative retrieval is thought to mediate value updating in this paradigm. We add citations here from the original Brogden paper and from a more recent review that nicely links these constructs together:

“Sensory preconditioning is a well-established paradigm in which subjects first form an association between two stimuli ('direct' or S_d_ and 'indirect' or S_i_) and then form an association between the direct stimulus and a reward (6). Generalization of value to the indirect stimulus is evidence of retrieving the learned association (24). Using fMRI, [96] showed that neural representations of the associated indirect stimulus are reinstated when direct stimuli are presented during the Reward-learning phase, and this retrieval is linked to the generalization of value from direct to indirect stimuli. This suggests that reinstatement through the learned associative link may be part of the mechanism for value updating. Our aim here is to explore the temporal structure of this reinstatement, which may help to shed light on the mechanisms of value updating as well as providing general insight into the dynamics of representations during retrieval.”

*3) The rationale for using univariate vs. multivariate feature sets should be stated earlier in the text. Is the rationale only to verify that more information is captured using multivariate techniques compared to univariate techniques? If so, I think simply citing the prior literature demonstrating this should be sufficient and it is unnecessary and a bit tangential to focus on this point as a primary finding. As the text stands now, it takes a while to get to the point of the primary research findings*.

Thank you for this excellent suggestion. We have moved the univariate classification analysis to the supplement, leaving just the following in the main text:

“As in many previous studies (cf. [8]; [61]), the extra sensitivity achieved by combining multiple features supported the use of multivariate analysis to track neural representations (Figure 3—figure supplement 1).”

*4) Pattern classification performance was calculated within subject and then averaged across subjects, revealing peaks around 200ms and 400ms. To what extent are these peaks evident in individual subjects? The authors partially raise this issue themselves in the Discussion section, but may wish to provide quantitative results regarding this issue*.

We have added a supplementary figure (new Figure 3—figure supplement 2) showing the raw decoding accuracy curves for individual subjects. Although the single-subject curves rest on very little data compared to the group curve, and so should be interpreted cautiously, some individual subjects may show a qualitative pattern of multiple peaks in decoding accuracy.

We also show betas from the quadratic term (a measure of positive curvature) of a regression of time on decoding accuracy between 200ms and 400ms, fit separately for each subject. No individual subject reached significant curvature (i.e. a significant beta on the quadratic term in the regression) after Bonferroni correction. This could be consistent with the idea that the group-level effect is being driven by a trend present in many subjects, rather than by a few extreme subjects.

5) Although the data generally supports the claim that they are primarily measuring “evoked” activity, the authors should be cautious to not imply that it is only evoked activity.

Given several of the reviewer comments, we realized it would take more figures and discussion to treat fully the time-frequency transformed analyses and the issue of evoked versus induced activity. As this issue is tangential to the conclusions of the current manuscript, we have decided to leave it for a future contribution. We therefore removed the supplementary figure about time-frequency analysis and any mention of evoked versus induced signals.

*6) In the Discussion section: Several EEG and fMRI studies of temporal order memory (some using an RSA approach) are excluded but should be cited and/or discussed (Hsieh, 2011, Journal of Neuroscience;*
[32]*, Neuron;*
[17]*, Neuron)*.

Thank you for pointing these out. We have added the citations with the following text:

“In humans, frontal theta power (33) and patterns of activity in hippocampus (17; 32) are implicated in coding temporal order within sequences of stimuli. Applying methods from the present work could be useful to establish a finer grained map of the representations used in complex memory and decision processes.”

*7) Overall, the data support the claim of two temporally distinct representational periods. However, such an interpretation is complicated by a highly significant (double the chance rate) classification performance during the two windows. The authors should discuss this more thoroughly. Additional analyses described above may also clarify this interpretational discrepancy*.

High classification performance within each window seems to suggest that there is some consistency in the pattern of activity evoked at this latency following stimulus onset. We have also added the following paragraph to the Discussion:

“Supporting the idea of these ∼200ms and ∼400ms epochs as distinct representational periods, we note the following. First, there was a decrease in classification accuracy between these periods. Second, classifiers trained on one epoch had low accuracy in the other epoch (Figure 3—figure supplement 5), suggesting information about the stimuli was coded differently between epochs. Third, the epochs had different similarity structure with respect to the stimulus categories (Figure 3). Fourth, the patterns from the two epochs were doubly dissociated in terms of their retrieval at S_d_ versus outcome (Figure 4), while the time period between the two peaks (i.e., around 300ms post-stimulus) was not strongly retrieved either at S_d_ or outcome (Figure 4).”

Reviewer 3:

*1) What kinds of multiple comparisons corrections are the authors using for the analyses shown in*
Figure 4*B? Elsewhere in the paper, the investigators appear to be using an approach of controlling for familywise error at p < .05, but it is not clear what is going on here (the paper only says p = 0.05 “peak-level significance thresholds”). Whatever correction the authors use will need to correct for the use of classifiers trained on different time points and also their exploration of multiple time points during the reward learning phase. For what it's worth, I thought that the approach taken in*
Figure 6
*(controlling for all time points, but only two classifiers: the 200ms-trained classifier and the 400ms-trained classifier) was acceptable; a similar approach could be applied here, if it isn't being used already*.

We actually used a more conservative correction for Figure 4. At each time point of training (i.e., the y-axis), we randomly permuted the category labels used to train the classifiers 1000 times. We then tested each of these null classifiers at each time point of testing (the x-axis). Thus, each of the 1000 permutations generated a time-by-time map of classification accuracies like those in Figure 4. We found the peak of each of these null accuracy maps, yielding 1000 peaks. The 95th percentile of these 1000 peaks is represented by the black contours in 4A and 4B. This statistical correction is especially powerful since it doesn't rest on any prior assumptions about there being interesting signals at 200ms and 400ms.

To make this clearer, we have added to the figure caption, and we have added the following text to the Methods:

“In Figure 4, we show 2-dimensional maps where the dimensions are times relative to two different events. To generate statistical significance thresholds for these maps, we recalculated these maps many times with independently shuffled category labels for the stimuli. Each shuffle yielded a map that contained no true information about the stimuli, but preserved overall smoothness and other statistical properties. The peak levels of each of these maps were extracted, and the distribution of these peak levels formed a nonparametric empirical null distribution. The 95th percentile of this distribution is reported as the significance threshold.”

*2) For the RSA analysis, it was not clear if the authors obtained a meaningful measure of “self-similarity” values (i.e., how similar is the pattern for a particular item to the pattern evoked by other instances of that same item). If they did not measure this, it would be useful to re-do the analysis in a way that obtains self-similarity measurements. Having this information would allow the authors to get separate readouts of item-specific information (e.g., by contrasting the diagonal cells to the off-diagonal cells from the same category) and category information. In particular, it would be useful to get these measures (along with some statistical assessment of their reliability, e.g., through bootstrapping) for both the 200ms and 400ms time points. While it is clear that category structure is not strongly represented at the 200ms time point, the RSA analysis in its current form does not speak to how strongly item-specific information is represented at the two time points (the text, as it is currently written, seems to be implying that there is less information about individual items at 400ms than 200ms, but we don't know that)*.

We agree. From the analysis we reported, we couldn't tell anything about the amount of information about individual items at 400ms vs. 200ms. The main claim we can make is that the neural representation of items within a category looks more similar at 400ms than 200ms, compared to between-category items. We have amended our descriptions of this throughout the paper, and excised all references to having less item-specific information at 400ms.

However, we pursued this suggestion directly by checking the similarity (measured by correlation) of neural responses evoked to different presentations of the same stimuli. These similarities were not different between 200ms and 400ms (p=0.3 by t-test). Thus, at least with this approach, we can't reach any conclusions about the absolute amount of item-specific information at 200ms versus 400ms. In the future, we think it will be interesting to investigate exactly what is different about the coding of the representations in different time epochs.

*3) As shown in*
Figure 1*, about half of the participants showed P(S*_*i*_*+) that was below .5, and some of these participants showed P(S*_*i*_*+) values that were well below .5. While I can understand how variance in the level of reactivation (during the reward learning phase) could lead to variance in participants' preference, ranging from no preference to strong preference in favor of the associated item, but it is not clear how variance in reactivation could lead to the opposite preference. The only other interpretation of the below-chance performance is that it is noise, but in that case, how can it be explained using classifier evidence? It would be useful if the authors commented on this*.

Very good point. We had been laboring under the misapprehension that having the wrong representation online could lead to negative behavioral preference. But in fact, as the reviewer noted, this doesn't make sense because you'd need to have the wrong ’category’ online to get below-chance decoding, and the wrong ’exemplar’ of a category online to get P(S_i_+)<0.5, and these two things are orthogonal. Thanks very much for bringing this to our attention.

In response to this, we split Figure 6 into two separate analyses: one using only subjects with P(S_i_+)>=0.5, and the other using only subjects with P(S_i_+)<=0.5. The new Figure 6 replaces Figure 6 from the original manuscript. It turns out that there is a much stronger correlation between behavior and decoding when we only consider P(S_i_+)>=0.5! Furthermore, the strongest peak of this correlation is at a time (∼400ms post-outcome) that might be easier to interpret than the peak in the original analysis (which was ∼100ms pre-outcome).

We would interpret this as validation of the reviewer's point: a positive behavioral preference can be driven by correct retrieval (i.e., above-chance decoding), but not vice versa. It's also nice that by cleaning up the analysis in this way, the effect is revealed to be highly significant (surviving nonparametric correction for peak level of the entire time-by-time map) rather than marginally significant.

We have also updated the Results and Discussion sections to reflect this new analysis.

*4) A suggestion: to the extent that classification performance is suffering due to a lack of training data, we have sometimes found in my lab that we can improve classification by using a “leave one subject out” approach (i.e., train on all but one subject, test on the left-out subject). This approach assumes that brain patterns are relatively consistent across participants. If that assumption is generally true, then the 25-fold increase in the number of training patterns can improve classification accuracy by a substantial margin (conversely: if there is extensive between-subject variability in the patterns, then moving to a leave-one-subject out approach can hurt classification accuracy). Anything that improves classification accuracy has the potential to greatly boost the interpretability of these results*.

Thanks for this suggestion. Early on we did experiment with training classifiers on multiple subjects' data together. In this data set it usually hurt performance more than it helped; and when we compared the patterns of individual subjects we found that they were often quite different from one another. We are very interested in exploring ways of pooling subjects' data together to improve classification, for example by learning some transformation to map subjects into a common space.
